# A Deep Learning-Based Robust Change Detection Approach for Very High Resolution Remotely Sensed Images with Multiple Features

**Lijun Huang [1,2], Ru An [1,\*], Shengyin Zhao [1], Tong Jiang [1] and Hao Hu [2]**

[1] School of Earth Science and Engineering, Hohai University, Nanjing 211100, China;
ljhuang@mail.ustc.edu.cn (L.H.); zhaosy51@hhu.edu.cn (S.Z.); jiangtong@hhu.edu.cn (T.J.)

[2] School of Mechanical and Electronic Engineering, Suzhou University, Suzhou 234000, China;
huhao@ahszu.edu.cn

\* Correspondence: anrunj@hhu.edu.cn

**Abstract:** Very high-resolution remote sensing change detection has always been an important research issue due to the registration error, robustness of the method, and monitoring accuracy, etc. This paper proposes a robust and more accurate approach of change detection (CD), and it is applied on a smaller experimental area, and then extended to a wider range. A feature space, including object features, Visual Geometry Group (VGG) depth features, and texture features, is constructed. The difference image is obtained by considering the contextual information in a radius scalable circular. This is to overcome the registration error caused by the rotation and shift of the instantaneous field of view and also to improve the reliability and robustness of the CD. To enhance the robustness of the U-Net model, the training dataset is constructed manually via various operations, such as blurring the image, increasing noise, and rotating the image. After this, the trained model is used to predict the experimental areas, which achieved 92.3% accuracy. The proposed method is compared with Support Vector Machine (SVM) and Siamese Network, and the check error rate dropped to 7.86%, while the Kappa increased to 0.8254. The results revealed that our method outperforms SVM and Siamese Network.

**Keywords:** change detection; deep learning; multiple features; radius scalable circular; very high-resolution remote sensing

## 1. Introduction

With the urban expansion of China, the technology of change detection (CD) in urban areas has become more and more important. Change detection techniques have great potential in the following five fields [1,2]: (1) Scene change detection, detection and analysis of land use changes at the semantic level; (2) Hyperspectral change detection, combined with the spectral mixing model, to analyze change type without supervision, and realize sub-pixel change detection; (3) Improvement of the method of classification change detection, making full use of the spatiotemporal correlation between multi-temporal images, which can improve consistency of results of multi-temporal classification and to improve accuracy of "from-to" change detection; (4) Multi-source and multi-resolution change detection, which study the theories and methods of common change detection and use of multi-temporal remote sensing data with different observation mechanisms and resolution; (5) Change Detection based on deep learning, where the spectral/spatial consistent features of multi-temporal images were extracted by deep learning neural networks, and the high precision results of change detection were obtained.

Change detection (CD) from multi-temporal remotely sensed images is one of the important technologies for information processing [1]. CD has significant applications in various fields [1–8],

such as land-use/cover change, deforestation, urban expansion, and disaster monitoring. It can also be used for the identification of targets, such as bridges, ports, and military bases. In recent years, with the enrichment of remote sensed image data sources, CD has developed rapidly, and various methods have shown good application effects and potential in different fields. However, no method can be applied to the vast majority of cases, and no detection algorithm is the most optimal. The development of robust algorithms has always been the research focus in the field of CD [1,2,9]. There are numerous studies on change detection. Du et al. (2012) proposed two methods of CD based on the information fusion strategy: (i) weighted similarity distance in one-dimensional feature space, (ii) fuzzy set theory and support vector machines (SVM) in multi-dimensional feature space [1]. Zhang et al. (2017) investigated the CD with multi-temporal remote sensing images [2], and introduced three aspects: pre-processing, thresholding, and accuracy assessment. Xiao et al. (2016) proposed a new framework combining pixel-level CD and object-level recognition [4]. Thonfeldet, F. et al. (2016) proposed a method called robust change vector analysis (RCVA) to reduce the errors generated by the sun's position and sensor viewing geometry [9]. However, RCVA sometimes cannot resolve the correspondence point-matching problem very well in a rotating image. Zhang et al. (2017) built and trained a model based on the Gaussian Bernoulli Deep Boltzmann Machine with a label layer to extract deep features [10]. They determined the changed and unchanged areas, considered the information of the neighborhood, and thus reduced the registration error. However, this method was still based on the pixel-level operation, and the result of CD was too fragmented to be satisfactory. K. Li et al. [11] proposed an asynchronous feature tracking method based on line segments with the dynamic and active-pixel vision sensors (DAVIS).

Our paper makes full use of the features, representing different information in multi-temporal images [12–16], and introduces deep learning to excavate the features that are not easy to discover [14]. The dataset, combining the features of pix–depth–object [17], is used to train our model and predict the experimental area and the extension of it. These multiple features can complement each other for CD. The feature of the variance of the experimental area image is extracted by the Gray Level Co-occurrence Matrix (GLCM). With the development of deep learning technology, various models, such as LeNet, AlexNet, ZFNet, and NiN [18,19], have been used in the processing of remotely sensed images. The depth feature in this paper was extracted by VGG (Visual Geometry Group) model. U-Net has been a remarkable and the most popular deep network architecture, and it is introduced into the change detection in this paper [18–32]. Neighborhood analysis is introduced to look for the corresponding image points of two-time images (potentially distorted images) and to find their difference [33–42]. The method of circle neighborhood analysis reduces the error of constructing the difference image, and improves the precision of the U-Net.

## 2. Methodology

This paper proposes a CD framework, as shown in Figure 1, which performs the following steps. First, it needs to complete some preprocessing, such as registration to reduce geometric errors and radiometric correction to reduce errors caused by the difference of solar angles in different images. This provides preprocessing work for CD. Second, a variety of methods are used to construct the feature space, including a depth feature based on VGG convolutional networks, a texture feature based on GLCM, and a segmented object feature. Third, the framework constructs the difference image combining the three features and considers spatial–contextual information in an adaptive circular neighborhood in order to search for the corresponding image points. This method can reduce the errors by the difference of the sun's position and rotation, shifting the image. Fourth, works for the deep learning model—the principal component analysis (PCA) is applied to the differential images of object features, VGG depth features, and texture features. The first of the principal components is extracted from the different images, and then they are stacked to a new difference image. The dataset used for training is built by segmenting the difference image for the U-Net model. To enhance the robustness of the model, the difference image is segmented into 1000 small pictures of 256 × 256 pixels,

which are used for other operations, such as rotation, adding noise, and blur. At the same time, it needs to build the corresponding training labels. The training set, including the small pictures and labels, is used to train the U-Net model. Fifth, the model is used to predict the changes in the experiment area and to extend it. Sixth, the results of this paper are compared with SVM and Siamese Network. Finally, the accuracy of the experimental results is evaluated.

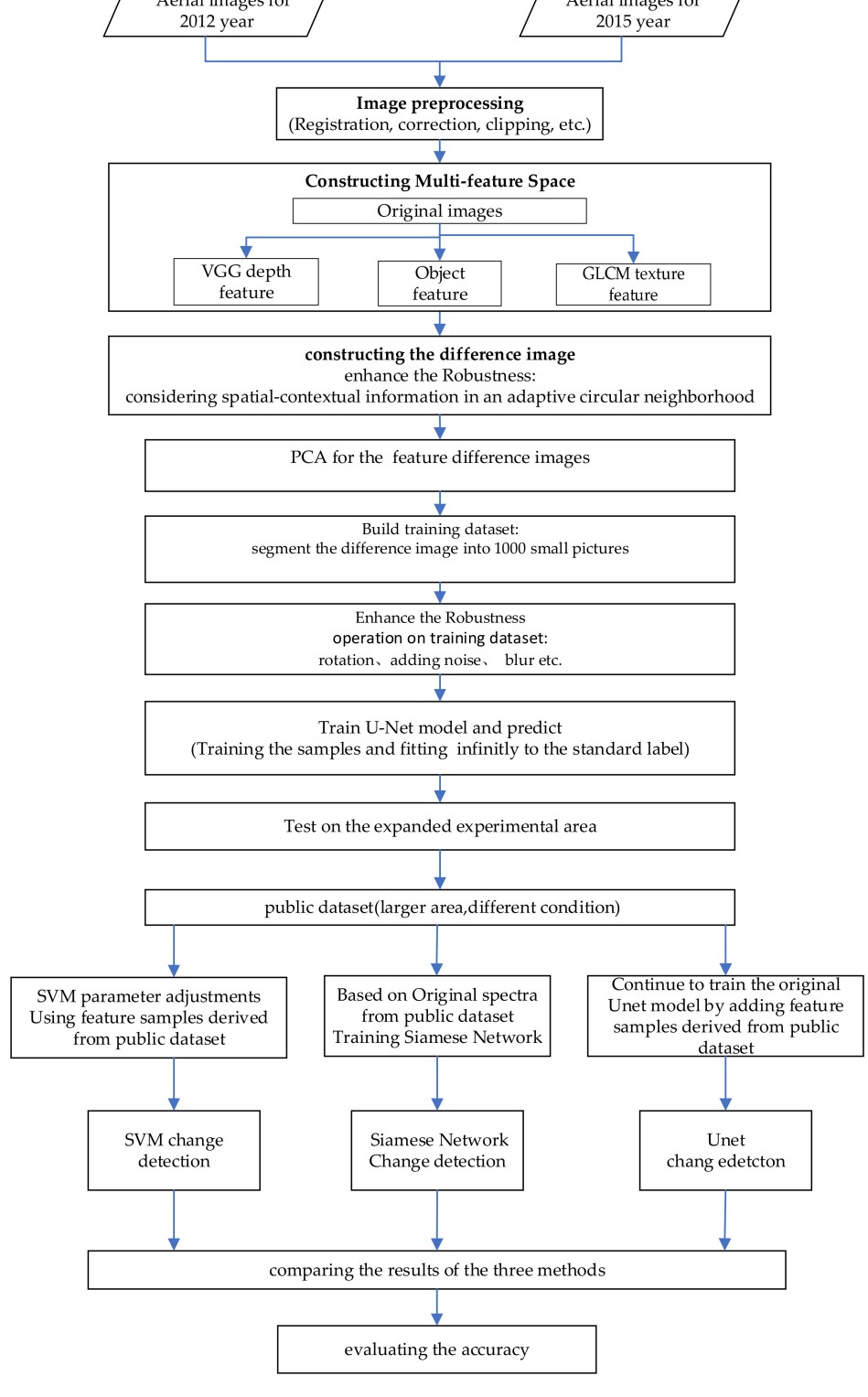

**Figure 1.** Flowchart of the proposed change detection framework.

### 2.1. Feature Space Construction

### 2.1.1. VGG Depth Feature

The VGG was developed by the Visual Geometry Group of Oxford University. The architecture of VGG is similar to AlexNet; however, VGG made some improvements based on AlexNet. The specific improvements are as follows: (1) the replacement of the larger convolution kernel ($11 \times 11$, $7 \times 7$, $5 \times 5$) in AlexNet with several consecutive convolution kernels ($3 \times 3$); (2) VGG has a very simple structure, which uses the same convolution kernel size ($3 \times 3$) and maximum pooling size ($2 \times 2$) for the entire network.

In this paper, the VGG-19 model is used to obtain a class of feature images. The texture details in the convolutional layer of the VGG-19 are strongly invariant. It has 16 convolutional layers and five pooling layers. According to the degree and size of the abstraction of features, it can be divided into four layers: the shallow layer, middle layer, sub-deep layer, and deep layer [17]. The convolutional layer deeply analyzes each piece sampled from the previous layer by the convolutional kernel to obtain the features with a higher degree of abstraction. The pooling layer can convert a high-resolution image to a lower one and reduce the parameters of the neural network. The VGG-19 selects the convolution kernel of $3 \times 3$ to extract the peripheral information of each pixel. Although it requires an increased number of iterations, the extracted features are more detailed and comprehensive. With the increase of the depth of the model structure, more abstract high-level semantic features can be extracted. This model has a good generalization ability on different datasets. The VGG-19 model trained in this paper extracts 36 features.

### 2.1.2. Object-Based Feature

The object-based feature of CD has certain advantages, which can improve the efficiency of detection and reduce the processing of trivial patches. Object-oriented image analysis has been applied in the interpretation of images sensed remotely since the 1970s. Ketting and Landgrebe, by considering the advantages of homogenous object extraction, proposed a segmentation algorithm called the Extraction and Classification of Homogenous Objects (ECHO) in 1976. ECHO makes the homogeneous pixel form objects of different sizes through image segmentation. The object-oriented image analysis method divides the remotely sensed images into different homogeneous objects, each of which has various attributes describing the spectrum, shape, structure, and texture. The spatial relationships among these objects, such as adjacency and inclusion, as well as similar inheritance relationships in object-oriented software development, are known. It analyzes the entity; not a single pixel. The entity is a meaningful image object composed of multiple pixels. The object-oriented image analysis adopts a rule of multi-level image segmentation to generate polygon objects with similar attributes at any scale. Here, the fuzzy mathematics method (Faber and Forstner, 1999) is used to obtain each attribute. The image object—the basic unit of information—is then extracted automatically.

In this paper, an edge-based segmentation algorithm is used for the four bands, and the full Lambda schedule algorithm is utilized to partially merge over-segmentation fragments. The vector and raster diagrams are generated as shown in Figure 5. The adjacent pixels of the image are aggregated into a whole as a block of spectral elements. The image is segmented and classified by using the spatial, textural, and spectral information of the high-resolution data. This mainly includes two steps: image object construction and object classification.

### 2.1.3. Texture Feature

The texture is one of the most important features, providing a large amount of information for image recognition and understanding. In [7], X. Pengfeng et al. claimed that the variance difference between different ground objects is the largest. They applied GLCM to the raw image of four bands to obtain variance features. The grayscale co-occurrence matrix and the value of texture features were calculated on the sub-image formed by each small window. Subsequently, the value of the texture

feature representing this window was assigned to the center pixel of the window to complete the texture feature calculation of the first small window. After this, the window was moved by one-pixel-step on the raw image to form another small window image, and the new co-occurrence matrix and the new value of texture features was calculated. By analogy, a matrix of textured eigenvalues was formed to be transformed into a texture feature image. Here, the size of the window should not be too large (e.g., $9 \times 9$) as it increases the blurring and coarseness of the texture. Thus, this paper selected the small window size of $3 \times 3$.

## 2.2. Constructing a Robust Difference Image

Due to the sun's position and sensor viewing geometry, the identical ground areas of comparison are not correctly represented by the corresponding pixels in the bi-temporal images, which results in geometric distortions and misregistration. Accordingly, the expansion of the spurious changes becomes more serious, especially for the rotation image. To resolve this issue, this paper improves the RCVA\CVA method by including the analysis of pixels with position $j, k$ in images $x_1(j, k)$ and $x_2(j, k)$ as well as the pixels in adjacent circular neighbors. The robust difference images are calculated by considering pixel circle neighborhood to subtract $t1$ from $t2$, and vice versa. The value of radius $r$ is adaptive. (Equations (1) and (2)):

$$x_{diffa}(j,k) = \min_{\sqrt{(p-j)^2+(q-k)^2} \leq r} \left\{ \sqrt{\sum_{i=1}^{n} (x_2^i(j,k) - (x_1^i(p,q)^2} \right\} \tag{1}$$

$$x_{diffb}(j,k) = \min_{\sqrt{(p-j)^2+(q-k)^2} \leq r} \left\{ \sqrt{\sum_{i=1}^{n} (x_1^i(j,k) - (x_2^i(p,q)^2} \right\} \tag{2}$$

We calculated the minimum spectral difference for the circular neighborhood in multiple bands. We found that the pixel $x_2$ in the circle with center $(j, k)$ and radius $r$ shows the least spectral variance to $x_1(j, k)$. The pixel $x_2$ contains most of the corresponding ground information of $x_1(j, k)$. We compute the lowest difference of each pixel in all bands in a moving circle window. In [9], only the square neighborhood is considered. On the other hand, our paper designed the circular neighborhood without direction, where $r$ is scalable. This method not only minimized the shift of the instantaneous field of view, but the rotation also. Figure 2 shows that the corresponding image points of the identical ground (roof Angle) in the two images were deviated. Even if the images are rotated by an angle, the method in our paper can still find the corresponding points in the circle neighborhood, enhancing the robustness.

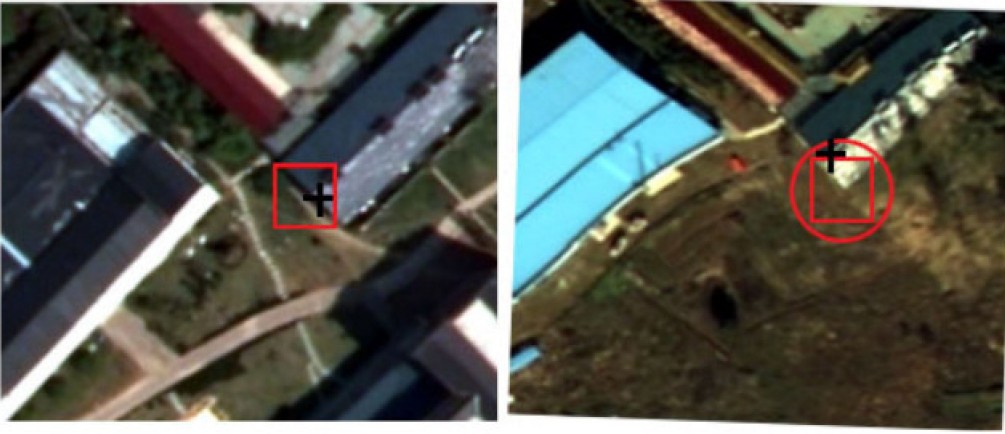

**Figure 2.** Position deviation of the identical ground object in the red box.

To improve the robustness of the difference image, we compare $x_{diffa}(j,k)$ with $x_{diffb}(j,k)$ and obtain the minor one as the difference image $M_{diff}(j,k)$:

$$M_{diff} = \begin{cases} x_{diffb}(j,k), x_{diffb} \leq x_{diffa}(j,k) \\ x_{diffa}(j,k), x_{diffa} \leq x_{diffb}(j,k) \end{cases} \tag{3}$$

In this way, the spectral change, by considering the circle neighborhood information, can be obtained.

### 2.3. Change Detection Based on U-Net

The U-Net is based on the expansion and modification of the convolutional network. It consists of two parts: (i) a contracting path to obtain context information, (ii) a symmetric expanding path for precise location estimation.

The U-Net can train good models on small datasets, similar to the one used in this paper. Furthermore, its training speed is very fast, providing satisfactory results in a short time.

The process for the U-Net is to cut, segment, rotate, add noise and other operations to expand the number of samples to segment the label image. This is followed by finding out the relationship between the training image and the corresponding label, and infinitely fitting to approximate this relationship. Finally, the parameters of the model are obtained. We use binary labels, i.e., change and no-change, and train a binary model. We make paddings for the pending image and fill an image with zeros, which is named image A. The size of the pending image is expanded to an integer multiple of 256, and then it is cut with 256 steps, resulting in a dataset consisting of $256 \times 256$ images. These images in the dataset are predicted by the model trained. After this, these predicted images are added to the location in image A. Finally, the new, larger image procured is cut into the size of the original image.

## 3. Experiment

In our experiment, we construct object features, depth VGG features, and GLCM-based variance texture features, all of which are derived from the aerial photographs of Yixing, Jiangsu Province, China, in 2012 and 2015. Using the method elaborated in Section 2.2, we constructed the difference images considering the circle neighborhood relation. SVM and Siamese Network were used to change detection based on the difference images. We constructed the training dataset to train the U-Net model and accordingly used the trained U-Net model to predict the test areas.

### 3.1. Data and Study Site

A portion of Yixing city, located in Jiangsu of China, is selected as the research area in this paper, and the longitude and latitude of Yixing City is $31°0 7 ′N - 31°37'$N, $119°31'$E$ - 120°03'E°$. The size of the experiment area is approximately 137 m $\times$ 107 m ($441 \times 346$ pixels), as shown the yellow frame in Figure 3. The size of the expanded experiment area is approximately 211 m $\times$ 203 m ($628 \times 678$ pixels), as shown the red frame in Figure 3. The two images in yellow frames are corresponded one by one, the same as the two images in the red frames. The two pairs of experiment sites are not very far from each other, so they have a lot of geographic similarities. They both are composed of several plants, roads, buildings, etc. The images for the experiment were acquired separately, in 18 February 2012 and 30 April 2015. The acquired images are orthotic. Subsequently, the different time-phase images are registered to make the registration error within one pixel. As seen in Figures 14 and 16, they are the ground truth images, which are obtained by visual interpretation, on-the-spot investigation and hand-drawn sketches.

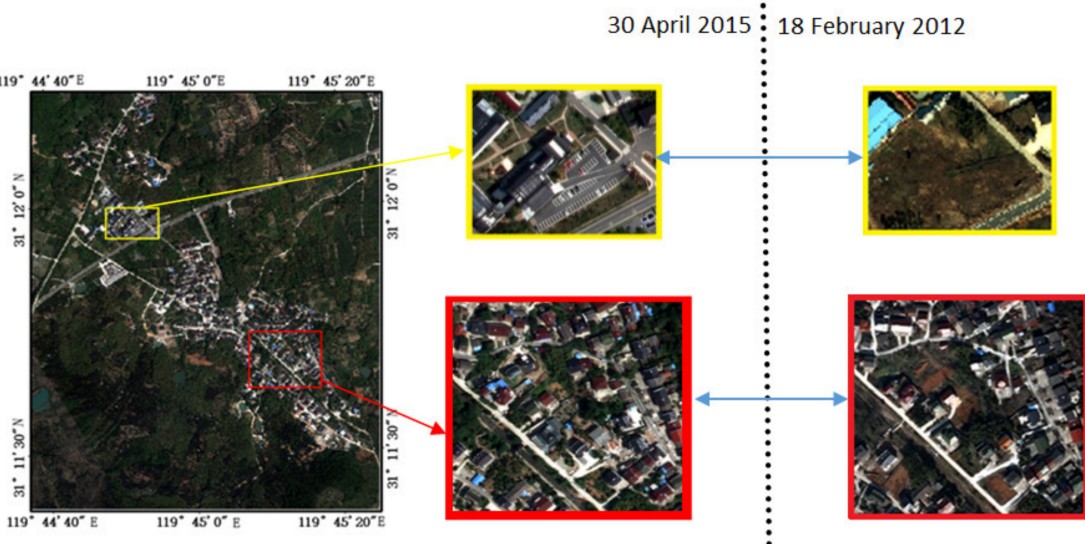

**Figure 3.** Experiment area.

The data acquired by the sensor UCXPWAG00315131 are used in this study, including four spectral bands: red, green, blue and near infrared. The spatial resolution is 0.31 m, and the gathering dates of the data were 18 February 2012 and 30 April 2015, respectively.

### *3.2. Feature Space Construction Experiment*

In this experiment, three methods were used to obtain object features, depth VGG features, and texture features respectively. The original spectra images are shown in Figure 4, which have four bands: R, G, B, and NIR. Figure 5a,b are respectively the segmented image of the experimental areas in 2012 and 2015, while (c) and (d) are the corresponding vector images. We adopt Edge algorithm to segment the image, and the scale level is 89.7. The Full Lambda Schedule algorithm is adopted to merge the fragments, merge level = 59.6. However, there are still too many fragmentary patches in segmentation, which requires the further combination of other features to detect the changes effectively.

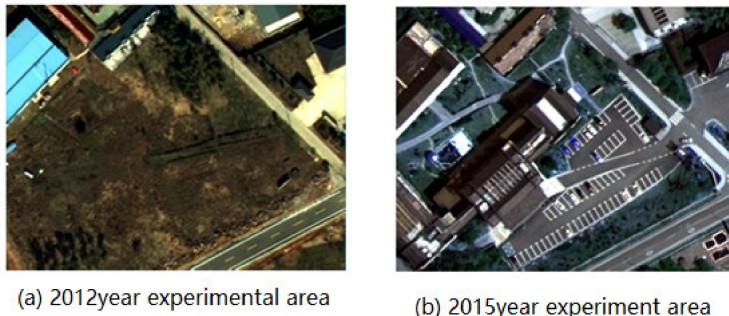

(a) 2012year experimental area     (b) 2015year experiment area

**Figure 4.** Original spectra image.

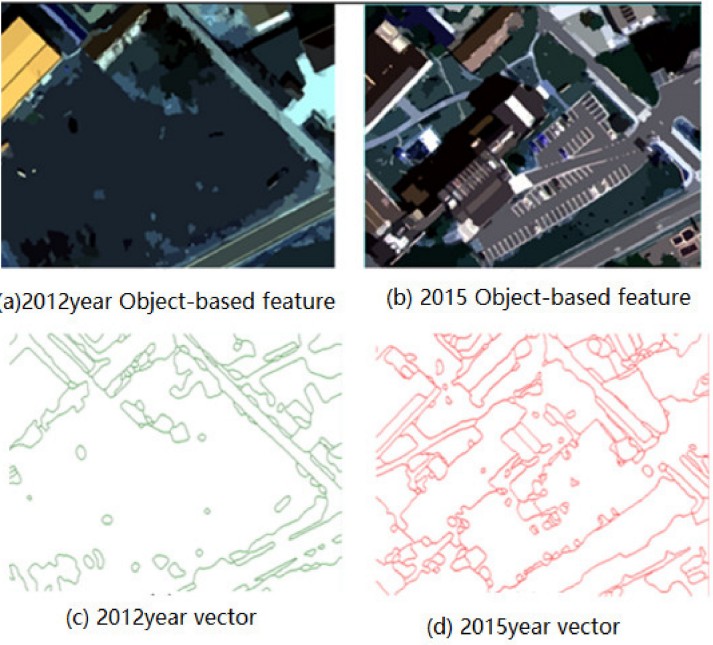

(a)2012year Object-based feature

(b) 2015 Object-based feature

(c) 2012year vector

(d) 2015year vector

**Figure 5.** Object-based features and corresponding vectors.

The VGG feature is based on pixels. A total of 72 features were derived by the VGG using the images taken in 2012 and 2015, i.e., 36 per each year. Five out of 36 VGG features are shown in Figure 6.

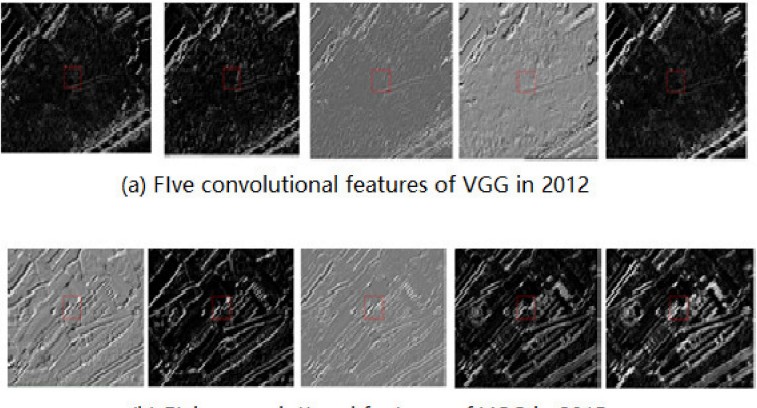

(a) Five convolutional features of VGG in 2012

(b) Fivie convolutional features of VGG in 2015

**Figure 6.** Convolution features of the Visual Geometry Group (VGG).

There are nine scalar parameters based on GLCM; mean, variance, homogeneity, contrast, to name a few. Our paper chooses the variance to derive the texture feature. In [3], by comparing the GLCM scalar parameters of different ground objects in high-resolution remote sensing images, P. Xiao et al. concluded that the variance differs the most. Considering this, we also chose GLCM variance to extract the feature. Figure 7a,b shows the texture feature images of 2012 and 2015, respectively.

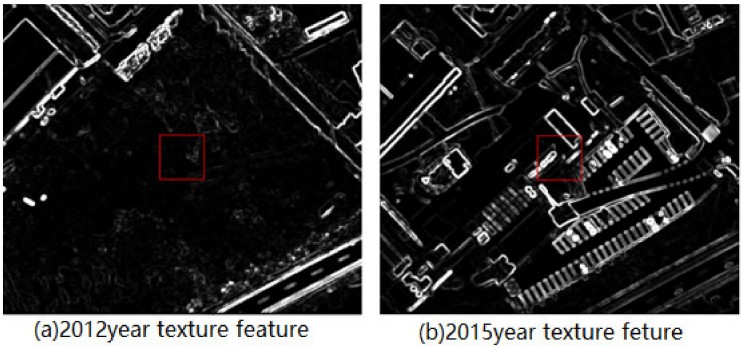

(a)2012year texture feature       (b)2015year texture feture

**Figure 7.** Texture feature.

### 3.3. Constructing the Difference Image

In this part of the experiment, principal component analysis (PCA) was adopted and performed on the feature space. The first component of the principal component analysis was obtained, which contains the main information as shown in Figure 8. In the Figure, (a), (c), and (e) are the first components of the object feature, the VGG feature, and the texture feature, respectively, in 2012, whereas (b), (d), and (f) are the first components of the object feature, the VGG feature, and the texture feature, respectively, in 2015.

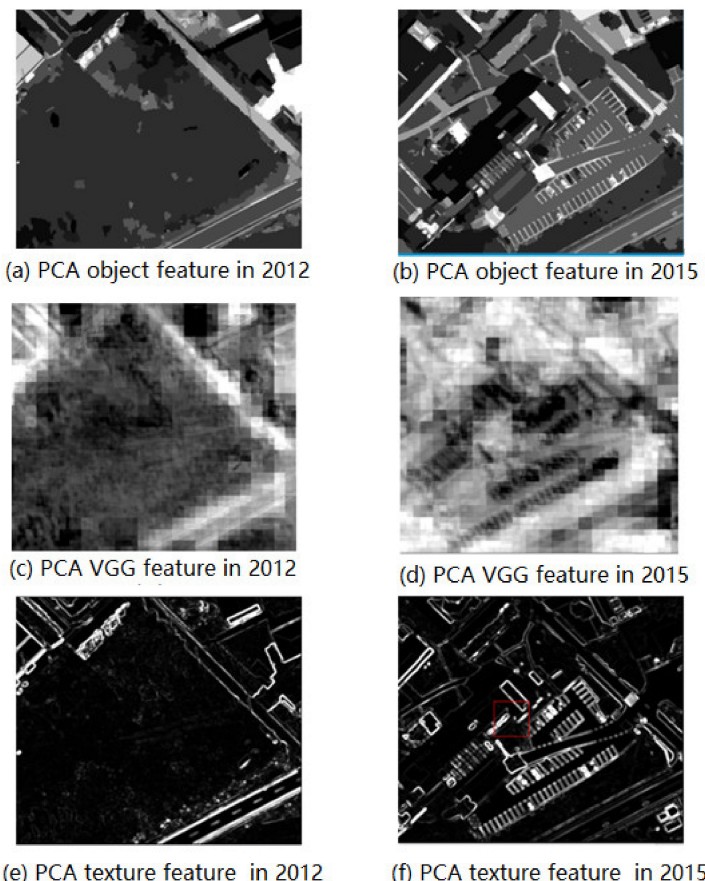

(a) PCA object feature in 2012      (b) PCA object feature in 2015

(c) PCA VGG feature in 2012      (d) PCA VGG feature in 2015

(e) PCA texture feature in 2012      (f) PCA texture feature in 2015

**Figure 8.** The principal components of features.

Afterwards, a difference image was created by considering a circular neighborhood with a scalable radius. Figure 9a is the difference image of the object, (b) is the texture difference image, (c) is the VGG difference image, and (d) is the RGB image integrating (a), (b), and (c). Here, R channel: object difference image, G channel: texture difference image, B channel: VGG difference image.

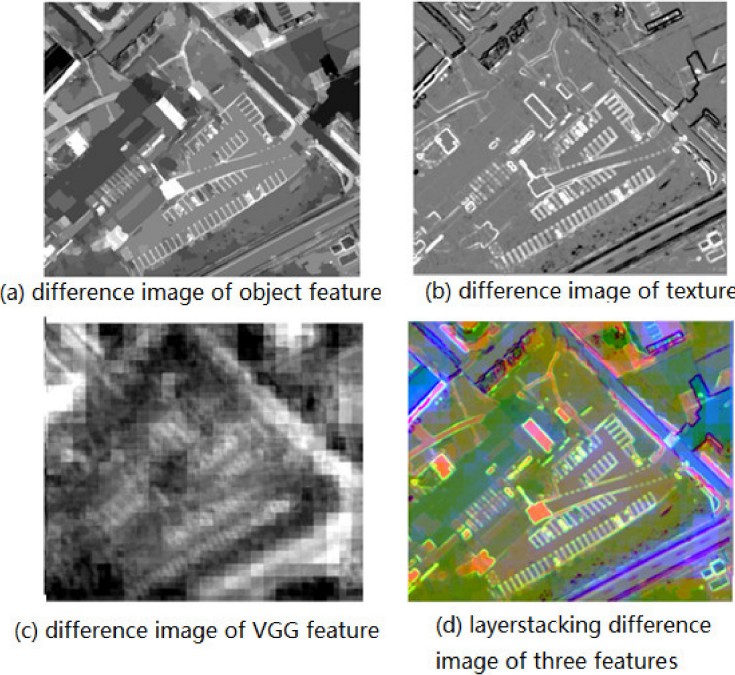

(a) difference image of object feature

(b) difference image of texture

(c) difference image of VGG feature

(d) layerstacking difference image of three features

**Figure 9.** Difference images.

### 3.4. Constructing a Robust Training Set for U-Net

The difference image, Figure 9d, was divided, rotated, noise-increased, blurred, etc. to obtain a rich training set. In this paper, 1000 samples were constructed in the dataset, all of which were small pictures of $256 \times 256$ pixels, parts of which as shown in Figure 10a. In addition, two representative pictures drawn from the 1000 samples are enlarged for a suitable display as shown in Figure 10b. In Figure 10b, the left one is one of the fuzzy rotation pictures, and the right one is the picture added with pepper and salt noises. Accordingly, the label training set was constructed. Moreover, the label training set and image training set were one-to-one corresponding, and the labels had 1000 samples. Figure 11 is the representative pictures drawn from the label training set, enlarged proportionally for clarity of display. In Figure 11a,b, the black represents the no-change part, while the white represents the change part. Figure 11 were corresponded to the Figure 10b, respectively.

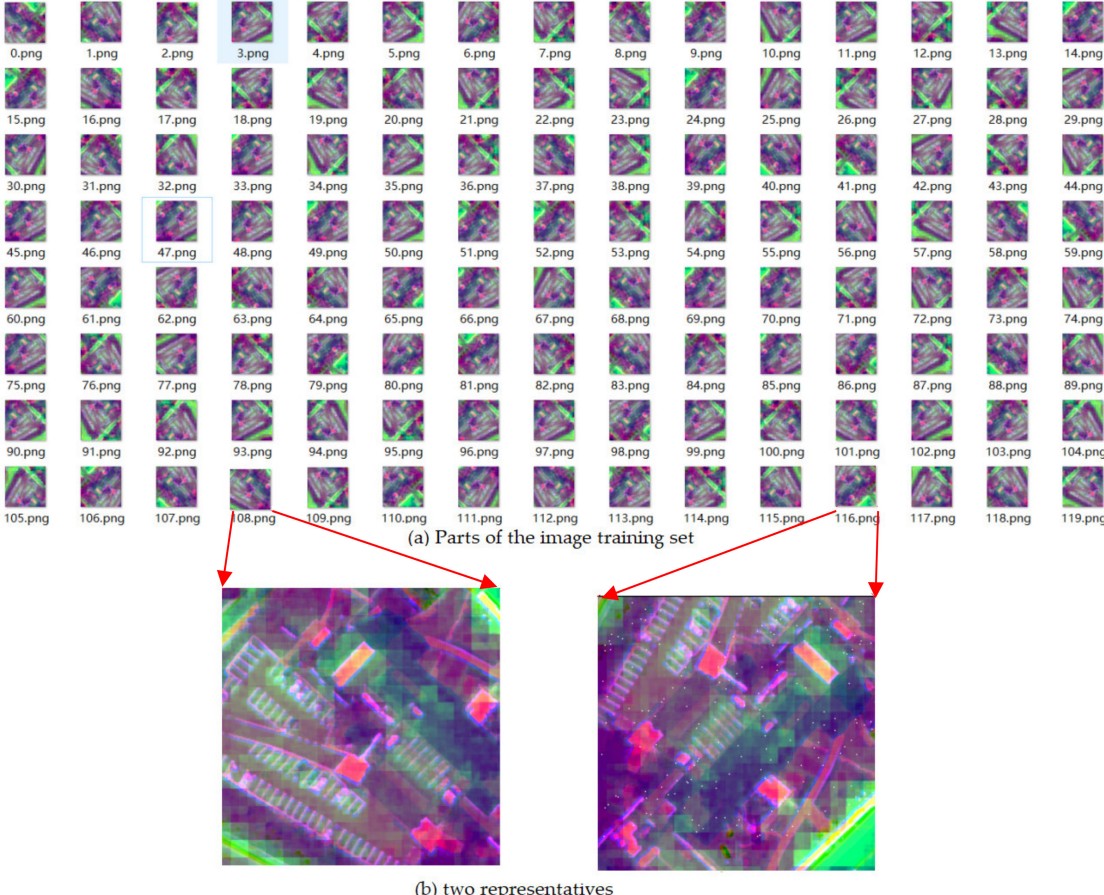

(a) Parts of the image training set

(b) two representatives

**Figure 10.** Training data.

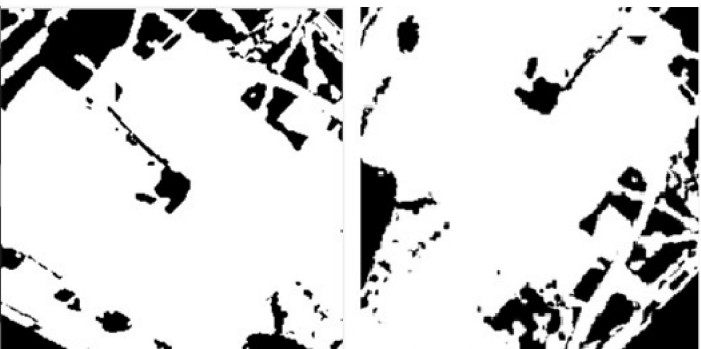

**Figure 11.** Two representatives in 1000 labels.

### 3.5. Model Training and Prediction Experiment

The U-Net is separately trained with 100, 150, 200 and 250 samples, and the models obtained are named as mod1-h5, mod2-h5, mod3-h5, and mod4-h5, respectively. As a representative shown in Figure 12, the training effect with 250 samples is displayed, and the training epoch is equal to 40, whereas the training and verification accuracies are close to 1. When the Epoch is within 0–10, the training and verification losses decrease rapidly, and stabilize when the Epoch exceeds 30.

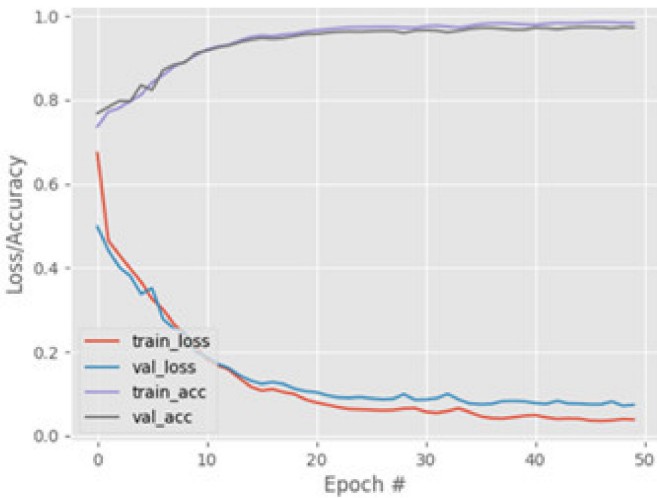

**Figure 12.** Track of loss/accuracy.

The four models trained were used for CD by binary prediction. The predicted results were provided in Figure 13, and the models achieved an accuracy of 61.3823%, 64.5825%, 73.2012%, and 92.3205% correspondingly compared to the truth ground model. The mod4-h5 showed the highest accuracy with Kappa coefficient of 0.8254. As a representative, mod4-h5's detailed pixel statistics and some relevant parameters are shown in Table 1. The truth ground image is shown in Figure 14.

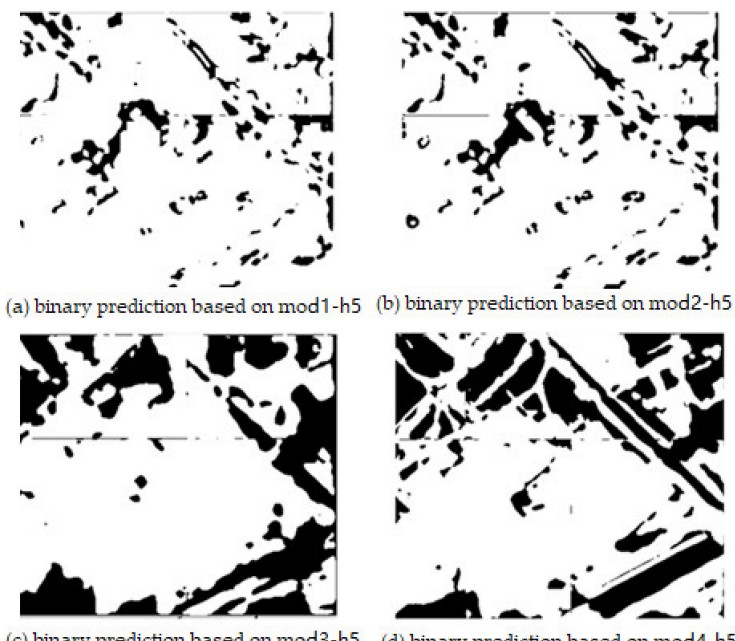

**Figure 13.** Predicted change image by four trained models.

**Table 1.** Evaluation parameters for Mod4-h5.

| Class | Commission (Percent) | Omission (Percent) | Commission (Pixels) | Omission (Pixels) | Overall Accuracy % | Kappa |
|---|---|---|---|---|---|---|
| No-change | 7.29 | 15.87 | 3437/47,172 | 8247/51,982 | 92.3205 | 0.8254 |
| Change | 7.86 | 3.43 | 8247/104,973 | 3437/100,163 | | |

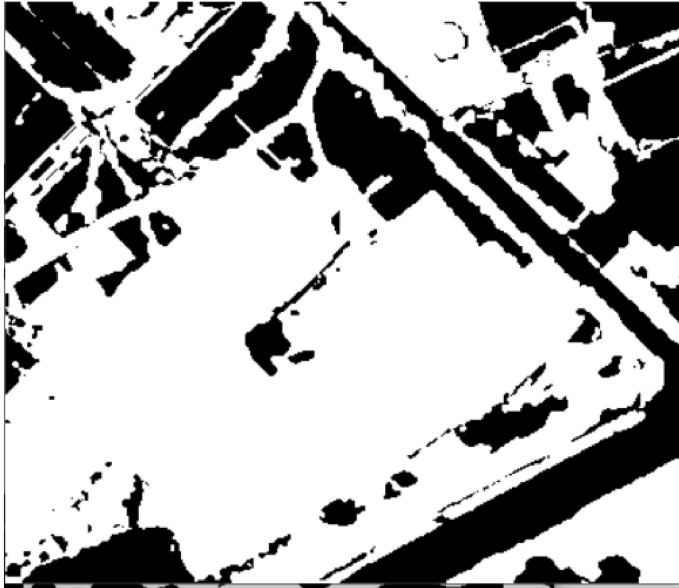

**Figure 14.** Truth round image.

The accuracy increases with the increase of the training data number. However, after 250 samples, this phenomenon lost its effect; thus, this paper uses 250 samples as the training data. We randomly chose data sets constructed in Section 3.4 as validation data, any samples in the 1000 samples without the 250 samples were used as training data.

*3.6. Model Working on the Expanded Experimental Area for Testing*

The expanded area predicted in this section is spatially not very far away from the experimental area in Section 3.5, and the scope is expanded to more than double to test the scalability of the trained model. If new parameters are trained with big training data for the model according to the process above sections of this paper, a higher accuracy will be obtained. However, this section applies to the trained model for the direct prediction of the expanded area without any additional work. This approach saves time, reduces costs, simplifies the process, and thus provides an idea for the extension of the method. Figure 15 shows the expanded experimental area, Figure 16 depicts the truth ground of change detection. Figure 17 is the prediction results of the expanded area using the model mod4-h5 trained in the previous sections.

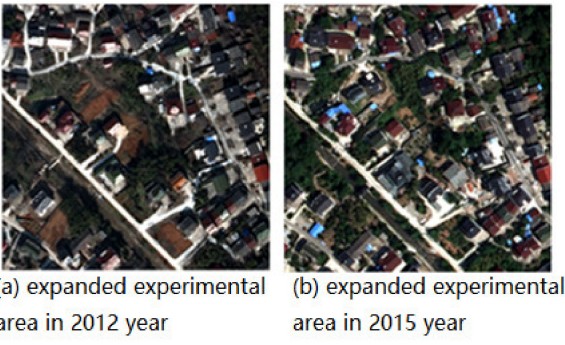

(a) expanded experimental area in 2012 year    (b) expanded experimental area in 2015 year

**Figure 15.** Expanded experimental area.

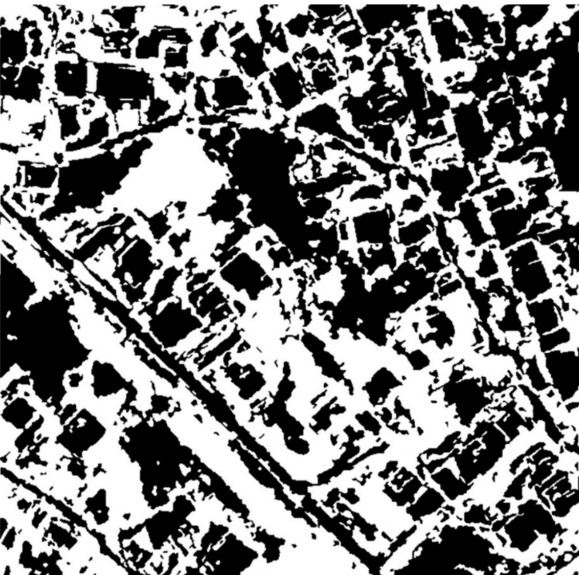

**Figure 16.** Truth ground of change in the expanded experimental area.

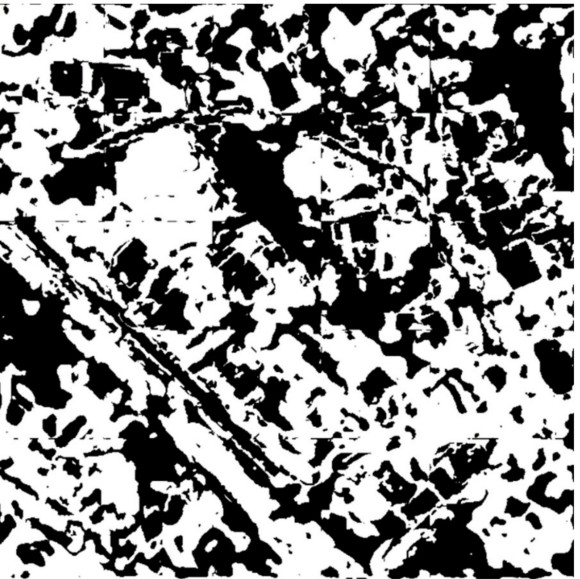

**Figure 17.** Prediction results.

The pixel statistics of change detection of the expanded area are shown in Table 2. The overall accuracy is relatively low, only 83.36%, as shown in Table 2. It can be seen that the model mod4-h5, trained in a small area, has certain limitations. If the region selected for training is more spatially related to the expanded experimental area, the similarity is higher, and the model obtained has more generalization ability and can predict the expanded areas more accurately. Although the overall accuracy of this section is only 83.36%, it greatly simplifies the detection process and improves the predicting efficiency.

**Table 2.** Result statistics for testing expanded area.

| Predict\Ground Truth (Pixels) | Change (Pixels) | No-Change (Pixels) | The Total (Pixels) | Overall Accuracy (Percent) | Commission (Percent) | Omission (Percent) | Kappa |
|---|---|---|---|---|---|---|---|
| Change | 195159 | 61894 | 257053 | | | | |
| No-change | 15066 | 190277 | 205343 | 83.36 | 24.08 | 7.17 | 0.67047 |
| Total | 210225 | 252171 | 462396 | | | | |

*3.7. Method Testing on Public Dataset Comparing with SVM and Siamese Network*

For the comparison of the proposed method with SVM and Siamese Network [43–45], the public dataset was downloaded from a website for testing. It was supplied by competition team "Sparse Characterization and Intelligent Analysis of Remote Sensing Images", sponsored by the Information Science Department of the National Natural Science Foundation of China. The feature space derived from the public dataset was constructed in Section 3.2, 3.3 and 3.4. The original spectra image from 2017, 2018, and the truth ground label image are shown in Figure 18.

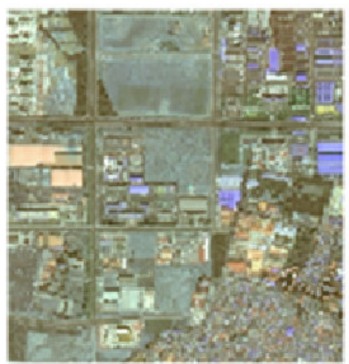 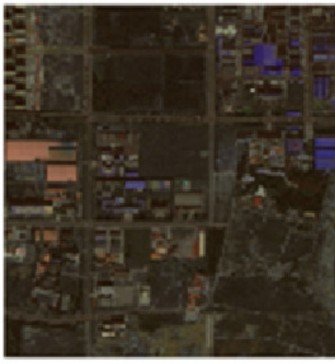 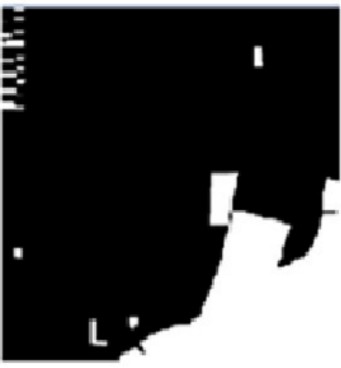

(a) 2017year original spectra image     (b) 2018year original spectra image     (c)2017year &2018year truth ground change label image

**Figure 18.** Public dataset.

The kernel type of SVM is Radial Basis Function. A total of 7979 sample points of change (white) and 3662 points of no-change (black) are selected by ROIs (region of interest) tools as training data for SVM. The binary classification is shown in Figure 19a, where the overall accuracy is 79.4449% (732164/921600) and Kappa coefficient is 0.5703.

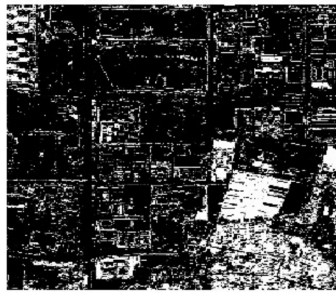 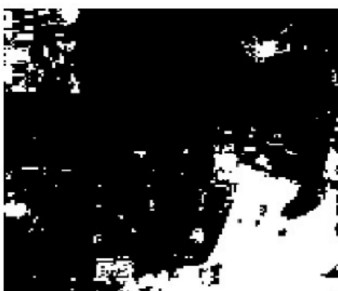 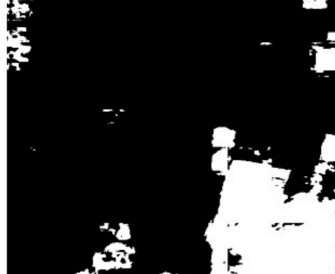

(a) binary change by SVM     (b) binarychang by Siamese Network     (c) binary by this paper method

**Figure 19.** Comparison of three change detection methods.

Figure 19b illustrates the binary changes by Siamese Network. First, Using the original spectral data from public dataset, we train the Siamese Net. After this, the change detection is carried out by the

network. Overall accuracy = (848,241/921,600) 92.04%, and Kappa coefficient = 0.7429. Figure 19c shows the results of our proposed method, which achieved and overall accuracy of 93.68% (863,354/921,600) and Kappa coefficient of 0.7634.

The comparison is shown in Table 3. The method proposed in this paper has the best overall accuracy of 93.68%, whereas the Siamese Network achieves 92.04%, and SVM performs the worst with 79.44% accuracy. Furthermore, our method performs best in terms of the check error rate (Commission 4.578%), and Kappa (0.7634). However, this paper's miss rate (omission 16%) is between the other methods' miss rates.

**Table 3.** Experimental results of three methods on the public dataset.

| Method | Overall Accuracy % | Omission % | Commission % | Kappa |
|---|---|---|---|---|
| SVM | 79.44 | 11.4 | 29 | 0.5703 |
| Siamese Network | 92.04 | 21.03 | 24.42 | 0.7429 |
| This paper's method | 93.68 | 16 | 4.578 | 0.7634 |

## 4. Discussion

Constructing feature space is very important for change detection. As one member in the feature space, the VGG feature is based on pixels, and it will produce "salt and pepper" in change detection. However, the VGG feature is not sensitive to the differences in light radiation, and this characteristic enhances the robustness of change detection, especially for the detection of changes in images with big radiation differences, as in the case of the two images shown in Figure 18a,b. Although the extracted object features provide considerable advantages for CD, they "equalize" the object interior, which causes the loss of some image details. To recreate these details, the pixel-based VGG features are utilized in this paper. The texture features can obtain the boundary information and contour of the ground objects and enhance their independence, as shown in Figure 5c,d. To overcome the position deviations of the ground objects, this paper proposes the circular neighborhood method, which can also reduce the errors caused by geometric registration and rotation, as shown in Figure 2.

The model working on the expanded experimental area in Section 3.6 performs slightly weaker and its overall accuracy on the expanded experimental area becomes from 92.32% to 83.36%, as the training set is too small. The accuracy can be improved if the training dataset is extended. Regardless, the accuracy of 83.36% essentially meets the demand. This is due to the good frame work we proposed and the robustness of the model U-Net, as shown in the flow in Figure 1. One important reason for this is the feature space constructed in Section 2.1, and another is the enhanced training dataset which is enhanced by operations such as increased noise, blurring, and cutting. All of the methods are adopted to improve the prediction accuracy of the model. In order to assess the effectiveness of this method on other datasets, we supplemented the public dataset for testing. In Section 3.7, for the comparison of the proposed method, we completed experiments on the public dataset using the method of SVM and Siamese Network. The SVM method is simple, however its accuracy is the lowest of all. The accuracy of the method we proposed in this paper is the highest (93.68%), while the commission is the lowest (4.578%). The Siamese Network performs slightly weaker than our method, however, it requires much more training data and has less robustness, and takes a lot of time to train the model.

## 5. Conclusions

This paper proposes a smart combination of different techniques to produce acceptable results. Due to the lack of training data, various methods (e.g., increased noise, blurring, and cutting) are adopted to enhance the adaptability of the U-Net model in complex situations, to improve its robustness. Therefore, the difficulty of lacking a sample set is solved. A novel adaptive neighborhood is introduced to reduce deviation while constructing a difference image.

However, the small-sized images of 256 × 256 are the results of the prediction mode, and they have to be spliced as large as the original image. As expected, the splice at the joint is filled with zeros (there is no detailed description in this paper). The splices, however, increased the error in change detection. Future studies should focus on reducing the error caused by splicing. In our paper, we focus on binary change detection (only two states: change and no-change) and we will pay more attention to multi-class change detection in future work.

**Author Contributions:** Conceptualization, L.H. and R.A.; methodology, L.H. and S.Z.; software, T.J. and S.Y.Z.; validation, S.Y.Z. and H.H.; formal analysis, L.H.; investigation, L.H.; resources, H.H.; data curation, L.H.; writing—original draft preparation, L.J.H.; writing—review and editing, R.A.; visualization, L.H.; supervision, R.A.; project administration, S.Z. and H.H.; funding acquisition, L.H. and R.A. All authors have read and agreed to the published version of the manuscript.

**Funding:** This research was funded by National Natural Science Foundation of China (Grant Number 41871326 and 41271361), the Jiangsu Province Key Research and Development Plan(BE2017115), Provincial Outstanding Young Talents Project of Anhui(2016XQNRL002).

**Acknowledgments:** We want to thanks Jiangsu Yitu Geographic Information Technology Co., Ltd. Provides the aerial photos of Yixing City, Jiangsu Province, China for our experiments.

**Conflicts of Interest:** The authors declare no conflicts of interest.

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
