# Peer review of "A Deep Learning-Based Robust Change Detection Approach for Very High Resolution Remotely Sensed Images with Multiple Features"

_remotesensing, doi:10.3390/rs12091441_

Round 1

Reviewer 1 Report

This manuscript proposes a change detection approach based on U-Net and multiple features including VGG depth feature, Object-based feature and Texture feature, and obtained a very high accuracy. I do not have critical comments on the methodology, only some small suggestions.

  1. Please add information about the ground truth, how it is obtained.
  2. Please be very careful about the index in the text. E.g. there are two sections indexed as 2.2.1, and the references are not correctly referred.
  3. Please add more information about the data rather than just give the sequence number of sensor UCXPWAG00315131, readers cannot find more information about it.
  4. Please give more information in the figure title, details need to be specified for subfigures.

Author Response

    your comments are all valuable and very helpful for improving our paper. We have studied comments carefully and have made correction which we hope meet with approval. Revised portion are marked by using red text. The revised words are shown in red style in the manuscript and also listed as follows:

  1. Please add information about the ground truth, how it is obtained.

  In the revised manuscript, we have added information about the ground truth in section3.1. 

    As seen in Fig.14 and Fig.16, they are the ground truth images, which are obtained by visual interpretation , on-the-spot investigation and hand-drawn sketches.

2.Please be very careful about the index in the text. E.g. there are two sections indexed as 2.2.1, and the references are not correctly referred.

    We have corrected the error index in the paper.

3.Please add more information about the data rather than just give the sequence number of sensor UCXPWAG00315131, readers cannot find more information about it.

   The two images in yellow frames are correspond one by one, and same as the two images in red frames. The two pairs of experiment sites are not very far from each other, so they have a lot of geographic similarities. They both are composed of several plants, roads, buildings, etc. The images for experiment are acquired separately in 18 February 2012 and 30 April 2015. The acquired images are orthotic. Then, the different time-phase images are registered to make the registration error within one pixel. As seen in Fig.14 and Fig.16, they are the ground truth images, which are obtained by visual interpretation , on-the-spot investigation and hand-drawn sketches.

4.Please give more information in the figure title, details need to be specified for subfigures.

    We have given more information in the figure title all of the paper, including subfigures.

Reviewer 2 Report

The method seems a smart combination of different techniques to produce acceptable results on a very small - for both number of samples and area extension - test set. Major work is required to make the paper suitable for publication. 

  • GLCM texture features do not provide accurate texture measures, especially when variance only is eventually considered in the final implementation
  • the test set should contain more examples and different scenarios in order to draw conclusions of general validity
  • the proposed method should be compared with state-of-the-art multi-class change detection methods
  • one of the high-performance competing methods to be considered should be, for example, "Learning Spectral-Spatial-Temporal Features via a Recurrent Convolutional Neural Network for Change Detection in Multispectral Imagery", IEEE TGRS Feb 2019.
  • OTSU method is not exactly a change detection method, but an automatic thresholding algorithm and should not be considered for testing.
  • the SVM implementation is certainly incorrect, since a Cohen's Kappa value of 0.32 is too low
  • I would suggest to check all implementations and reconsider the experimental phase.

Author Response

    your comments are all valuable and very helpful for improving our paper. We have studied comments carefully and have made correction which we hope meet with approval. Revised portion are marked by using red text. The revised words are shown in red style in the manuscript and also listed as follows:

1.GLCM texture features do not provide accurate texture measures, especially when variance only is eventually considered in the final implementation.

    Maybe GLCM texture features do not provide accurate texture measures, however the variance is eventually considered in our implementation. Because practice has proved it feasible. Professor Xiao, P has done experiment to compare the variance with contrast, dissimilarity, homogeneity, angular second moment, energy, entropy, mean, variance, and correlation. To select effective textural measures for built-up land change detection, they had tested three types of subsets of high-spatial resolution images, which represented factory built-up land, residential built-up land, and cropland, respectively.

    The References [4] “Xiao, P.; Wang, X.; Feng, X.; Zhang, X.; Yang, Y. Detecting China's Urban Expansion Over the Past Three Decades Using Nighttime Light Data. IEEE J.-Stars. 2014, 7, 4095-106”

See In the References [4] section 2.1.1,“ The variance measure showed largest difference in the mean values between built-up land images and cropland image than the other measures. Thus, the GLCM variance measure was selected as the textural feature for change detection.

2.the test set should contain more examples and different scenarios in order to draw conclusions of general validity

    In the revised manuscript, we have supplemented public dataset for testing and redo the experiments. See section 3.7 of the paper. The dataset was supplied by competition team "Sparse Characterization and Intelligent Analysis of Remote Sensing Images" sponsored by the Information Science Department of the National Natural Science Foundation of China.

3.the proposed method should be compared with state-of-the-art multi-class change detection methods

    However we focus on binary change detection in this paper. Only two states: change & no-change. We will pay more attention to multi - class change detection next step.

4.one of the high-performance competing methods to be considered should be, for example, "[1]Learning Spectral-Spatial-Temporal Features via a Recurrent Convolutional Neural Network for Change Detection in Multispectral Imagery", IEEE TGRS Feb 2019.

  "Learning Spectral-Spatial-Temporal Features via a Recurrent Convolutional Neural Network for Change Detection in Multispectral Imagery", IEEE TGRS Feb 2019.” The reference are valuable and very helpful. This ReCNN maybe competing, however I am inclined to use Siamese network by comparing in revised manuscript

5.OTSU method is not exactly a change detection method, but an automatic thresholding algorithm and should not be considered for testing.

    Replace it by Siamese network

6.the SVM implementation is certainly incorrect, since a Cohen's Kappa value of 0.32 is too low

   In revised manuscript, we replaced dataset and redo SVM experiment, Kappa value is 0.5703.

7.I would suggest to check all implementations and reconsider the experimental phase.

    We check carefully the implementations of three methods, and redo the experiments. See in sectin3.7.

Reviewer 3 Report

In the paper it is proposed a new approach for Change Detection (CD) for high resolution remotely sensed images, which is applied on a smaller experimental area, and then extended to a wider range area. The authors design a feature space including object features, VGG (Visual Geometry Group) depth features, and texture features. The trained deep model is used by the authors to predict many experimental areas, and they obtained high accuracy (92.3%). The proposed method is compared with SVM and OTSU (Maximum Between-Class Variance), and the results revealed that the proposed method outperforms both SVM and OTSU.

The overall quality of the paper is good, even if the authors must address the following two comments before a possible publication:

  • The introduction is too short and an overview of the state of the art is missing. I recommend the authors to improve the introduction and to better report the related works which face the same problem. Even a better introduction and description of the CD problem is needed.

  • The proposed experiments show that the method outperforms OTSU and SVM, but there is no link with the present literature. No comparison is provided to understand if the proposed method is better with respect to the other mentioned CD methods. Further, the method is tested only on the data provided by the authors, which makes it even more difficult to assess if the proposed CD method is better with respect to the state of the art. The authors must provide a better comparison with the other state of the art methods to understand this point.

Author Response

your comments are all valuable and very helpful for improving our paper. We have studied comments carefully and have made correction which we hope meet with approval. Revised portion are marked by using red text. The revised words are shown in red style in the manuscript and also listed as follows:

1.The introduction is too short and an overview of the state of the art is missing. I recommend the authors to improve the introduction and to better report the related works which face the same problem. Even a better introduction and description of the CD problem is needed.

    We improved the introduction, and added the latest research on five aspects of change detection. Seen in the introduction of the revised manuscript. 

    2.The proposed experiments show that the method outperforms OTSU and SVM, but there is no link with the present literature. No comparison is provided to understand if the proposed method is better with respect to the other mentioned CD methods. Further, the method is tested only on the data provided by the authors, which makes it even more difficult to assess if the proposed CD method is better with respect to the state of the art. The authors must provide a better comparison with the other state of the art methods to understand this point.

    We supplemented with the corresponding references [44,45], however we replace the method OTSU. Then we introduced the state-of-the-art Siamese network for change detection comparing with our method. In the revised manuscript, our method not only test on the data provided by us, but also tested on the public dataset that we supplement.

The Additional references:

[43] Mou, L.; Bruzzone, L.; Zhu, X.X. Learning Spectral-Spatial-Temporal Features via a Recurrent Convolutional Neural Network for Change Detection in Multispectral Imagery. IEEE T. Geosci. Remote. 2019, 57, 924-35.

[44] Dong, Y.; Wang, F. Change Detection of Remote Sensing Imagery Supported by KCCA and SVM Algorithms. Remote Sensing Information. 2019, 34, 144-8.

[45] Dunnhofer, M.; Antico, M.; Sasazawa, F.; Takeda, Y.; Camps, S.; Martinel, N.; Micheloni, C.; Carneiro, G.; Fontanarosa, D. Siam-U-Net: encoder-decoder siamese network for knee cartilage tracking in ultrasound images. Med. Image Anal. 2019, 60, 101631.

Round 2

Reviewer 3 Report

I am satisfied by the new version provided by the authors. They addressed all my concerns.